# Transfer Learning for Wireless Fingerprinting Localization Based on Optimal Transport

**DOI:** 10.3390/s20236994

**Published:** 2020-12-07

**Authors:** Siqi Bai, Yongjie Luo, Qun Wan

**Affiliations:** School of Information and Communication Engineering, University of Electronic Science and Technology of China, No.2006, Xiyuan Ave, West Hi-Tech Zone, Chengdu 611731, China; 201511020304@std.uestc.edu.cn (S.B.); luoyongjie@cdu.edu.cn (Y.L.)

**Keywords:** fingerprinting localization, transfer learning, optimal transport, indoor positioning, adaptive radio map

## Abstract

Wireless fingerprinting localization (FL) systems identify locations by building radio fingerprint maps, aiming to provide satisfactory location solutions for the complex environment. However, the radio map is easy to change, and the cost of building a new one is high. One research focus is to transfer knowledge from the old radio maps to a new one. Feature-based transfer learning methods help by mapping the source fingerprint and the target fingerprint to a common hidden domain, then minimize the maximum mean difference (MMD) distance between the empirical distributions in the latent domain. In this paper, the optimal transport (OT)-based transfer learning is adopted to directly map the fingerprint from the source domain to the target domain by minimizing the Wasserstein distance so that the data distribution of the two domains can be better matched and the positioning performance in the target domain is improved. Two channel-models are used to simulate the transfer scenarios, and the public measured data test further verifies that the transfer learning based on OT has better accuracy and performance when the radio map changes in FL, indicating the importance of the method in this field.

## 1. Introduction

In the mode of pervasive computing, people can acquire and process information at any time, any place, and in any way. Location information is essential for pervasive computing. Satellite positioning technology has been able to meet most outdoor location acquisition requirements, and indoor positioning technologies are constantly emerging to get through the “last meter” of positioning technology [1]. According to wireless technologies, the existing methods include Wifi positioning, Bluetooth positioning, ultra-wideband positioning, and lidar positioning, and so on. According to the measurement techniques, the existing methods include time of arrival (TOA) positioning, angle of arrival (AOA) positioning, received signal strength (RSS) positioning [2], Channel State Information (CSI) positioning [3], etc. According to the algorithms, the existing methods include triangulation, direct positioning, fingerprint localization (FL), and so on. Compared to other techniques, the FL can be applied to a complex environment, with wide application scope and easy implementation [2,3,4,5].

FL avoids the need for artificial modeling of complex indoor wireless channels and is typically achieved through machine learning techniques such as classification or regression algorithms. As with traditional machine learning applications, FL usually assumes that the training fingerprint data (also called radio map) are sampled from the same distribution as the test data. However, in practice, many factors will cause fingerprint distribution to change. For example, when using RSS fingerprints as features, changes in Access Point (AP) number/location, positioning environment, or equipment parameters will all lead to changes in channel parameters, resulting in changes in the location fingerprint distribution. As a result, the model’s accuracy obtained from the old training data will decrease or even fail. This can be addressed by seeking transfer learning techniques.

Transfer learning studies how to transfer the knowledge from the old data domain to the new data domain to deal with the problem that there is no labeled data (unsupervised) or insufficient labeled data (semi-supervised) in the new domain. Transfer learning is a new paradigm in machine learning technology, which has been successfully applied in many machine learning fields, such as the biomedical field [6,7], language and text recognition field [8], graph neural network field [9], detection field [10], etc., and also in positioning [11].

The existing studies on FL transfer learning mainly consider three scenarios: time transfer [12,13,14], device transfer [15,16], and space transfer [17]. Time transfer refers to transferring the collected fingerprints or trained model knowledge from one time to another. Device transfer is the transfer from one device to another. Spatial transfer refers to the transfer from one area to another. No matter which kind of transfer is used, the essential learning is that the fingerprint distribution has changed from one state to another. Feature-based transfer learning has recently been verified to be effective in FL problems [18]. However, the frequently used maximum mean difference (MMD) distance does not consider the detailed differences between the two distributions.

In this paper, a novel transfer learning method based on optimal transport (OT) [19] is applied to FL. By introducing the Laplacian regularization and jointly learning mechanism, a smoother mapping function can be learned to improve the algorithm’s robustness further. We use both the free space channel model and the multi-wall model to simulate the proposed method’s performance and analyze the reason why the OT-based transfer learning performance is good. The performance of the algorithm is further verified on the public measured data set. The results indicate that OT technology is significant to the transfer learning problem in FL.

## 2. Related Works

### 2.1. Wireless Fingerprinting Localization

RADAR is one of the earliest wireless fingerprint positioning systems [20]. It adopts the KNN regression positioning method: a classic FL method with a simple algorithm and good robustness. Subsequently, support vector machine [21], decision tree [22], neural network [23], and other machine learning methods are also used in fingerprint positioning. These methods are deterministic methods, which map the location fingerprint to a specific location. Another type of method is called the probability method, which considers the position fingerprint’s randomness and maps a fingerprint into the probability density of the position. The usual methods are naive Bayes [24], probability kernel regression [25], conditional random field [26], and so on.

### 2.2. Transfer Learning in the Wireless Fingerprinting Localization

The research on the transfer learning of wireless FL started at about the same time as the general transfer learning. Yin et al. proposed a time transfer method of radio fingerprint map in 2005 [12], which uses regression analysis to learn the predictive relationship between the received signal strength of mobile devices and the received signal strength of reference points on sparse locations, to update the radio map in real-time according to the predictive relationship. LeManCoR [13] is a kind of adaptive method for fingerprint mapping based on the Manifold co-regularization. The LuMA method [15] takes into account both time transfer and device transfer. In addition, there are studies that consider multi-device transfer [16] and spatial transfer [17] in fingerprint positioning. All the above methods require the target domain to have a small amount of labeled data, so they belong to semi-supervised transfer learning. Furthermore, they all adopt the manifold alignment algorithm. TrHMM [14] is also a time transfer learning method for FL. Unlike the previous method, it is a parameter transfer learning method based on the Hidden Markov model. None of these methods adopt the latest available transfer learning algorithm.

General feature-based transfer learning methods include Transfer Component Analysis (TCA) [27], Joint Distribution Adaptation (JDA) [28], Balanced boundary Distribution Adaptation (BDA) [29], etc. These methods are types of unsupervised transfer learning, where there is no label information in the target domain. An improved method based on the general feature-based transfer learning has been successfully applied to FL recently [18]. However, the method has an excess of super parameters, and the performance may worsen when the super parameters are not chosen carefully. It is urgent to develop a fingerprint transfer learning method with fewer super parameters and robust performance in practical application.

### 2.3. Optimal Transport

The theory of optimal transport (OT) originates from what mathematician Gaspard Monge described in 1746 to 1818 as the “Monge problem” about how to move a sand pile to another place and change its shape into a predefined one at minimal cost [19]. It is then used by mathematicians to compare the distance between two probability distributions. In recent years, thanks to approximate solvers’ appearance that can be extended to high-dimensional problems, the revolution of OT technology has been initiated. It has been successfully applied to a variety of problems in image science (such as color or texture processing), graphics (for shape processing), or machine learning (for regression, classification, and generation modeling), as well as solving the problem of transfer learning [30].

## 3. Problem Description

FL is a method of associating location information with its fingerprint and then using parameterized or nonparametric models for location identification. Specific environmental features are the basis for creating fingerprint information. In wireless FL, it uses the wireless signal features to create a fingerprint of the position, also known as a wireless fingerprint. The fingerprints in the interesting area construct a radio map. Various wireless channel measurements can be selected as site-specific signal features, such as TOA, AOA, RSS, CSI, etc. Sometimes, they can be fused to form a higher-dimensional feature space. The signal feature is mapped to the position fingerprint in a preset way. Then, a sample associated with position u can be expressed as $x\left(u\right)=\left(x_{1}\left(u\right),\ldots ,x_{m}\left(u\right)\right)\in X^{m}$, and its conditional probability density is denoted as $P_{X\left|U\right.}$, where, $X^{m}$ represents the m-dimensional fingerprint space.

Wireless FL usually includes two stages: the offline stage and the online stage. A radio map is first created in the offline phase, which contains data pairs of several coordinates and corresponding fingerprints within the location area. Then, the radio map and the learning algorithm are used to get the position recognition function g:x→u, which maps a fingerprint to its estimated coordinate. In the online phase, the target’s estimated coordinate $g(\tilde{x})$ is obtained according to the online fingerprint $\tilde{x}$ and the function *g*.

In transfer learning of FL, it is assumed the data distribution in the offline and online phases comes from different distributions. Mathematically, let the source domain be $D_{s}=\left\{X_{s}^{m},P_{X}^{s}\left(x\right)\right\}$ and the task in the source domain be $T_{s}=\left\{U_{s}^{d},g_{s}\left(\cdot \right)\right\}$, where the subscript or superscript “*s*” represents the source domain, and if changed to “*t*”, it represents the target domain. The following text will follow this notation. $X^{m}$ represents the *m*-dimensional fingerprint space, and $U^{d}$ represents the *d*-dimensional position space. In this paper, d=2 is setted, and the superscripts “*m*” and “*d*” will be ignored below when it is well defined. g(·) stands for the position recognition function. The joint distribution of the offline phase, $P_{X,U}^{s}\left(x,u\right)$, is associated with the source domain. Moreover, the joint distribution of the online phase, $P_{X,U}^{t}\left(x,u\right)$, is associated with the target domain. In transfer learning setting, $P_{X,U}^{s}\left(x,u\right)\neq P_{X,U}^{t}\left(x,u\right)$. Transfer learning of FL studies how to transfer the knowledge about location from the old radio map (in the source domain) to the new one (in the target domain), so as to make full use of the knowledge of the source domain to optimize the task of FL in the target domain.

In practice, multiple instances of fingerprints in a domain can be observed, either with or without a location label. Assume that the fingerprint–coordinate pair set $D_{s}=\left\{\left(x_{i}^{\left(s\right)},u_{i}^{\left(s\right)}\right)\left|x_{i}^{\left(s\right)}\in X_{s},u_{i}^{\left(s\right)}\in U_{s},i=1,\ldots ,n_{s}\right.\right\}$ was observed in the source domain, including $n_{s}$ samples in total. In the target domain, the unlabeled fingerprint set $D_{t}=\left\{x_{i}^{\left(t\right)}\left|x_{i}^{\left(t\right)}\in X_{t},i=n_{s}+1,\ldots ,n_{s}+n_{t}\right.\right\}$ and labeled fingerprint set ${D^{\prime }}_{t}=\left\{\left(x_{i}^{\left(t\right)},u_{i}^{\left(t\right)}\right)\left|x_{i}^{\left(t\right)}\in X_{t},u_{i}^{\left(t\right)}\in U_{t},i=n_{s}+n_{t}+1,\ldots ,n_{s}+n_{t}+{n^{\prime }}_{t}\right.\right\}$ were observed, including $n_{t}$ and $n^{\prime }{}_{t}$ samples, respectively, and in general we have ${n^{\prime }}_{t}\ll n_{s}\ll n_{t}$. When ${n^{\prime }}_{t}=0$, it is called unsupervised transfer learning; otherwise, it is called semi-supervised transfer learning.

The transfer learning task of FL is to estimate the position recognition function ${\hat{g}}_{t}\left(x;D_{s},D_{t},{D^{\prime }}_{t}\right)$ in the target domain according to the observed fingerprint samples in the source domain and the target domain, so as to minimize the generalization error of the target domain. The generalization error is expressed as follows,
(1)$$E_{x\in X_{t}}\left[∥{\hat{g}}_{t}\left(x;D_{s},D_{t},{D^{\prime }}_{t}\right)-g_{t}\left(x\right)∥\right].$$

When $X_{s}=X_{t}$, it is called homogeneous transfer learning. This paper considers unsupervised homogeneous transfer learning.

Hypotheses should be made to theoretically guarantee the transfer learning to succeed [31]. The following are hypotheses often used in the transfer learning problem.
Class imbalance hypothesis: the distribution of labels in the two fields is different, i.e., $P_{Y}^{s}\left(y\right)\neq P_{Y}^{t}\left(y\right)$, but the conditional probability distribution of the feature is the same, i.e., $P_{X\left|Y\right.}^{s}\left(x\left|y\right.\right)=P_{X\left|Y\right.}^{t}\left(x\left|y\right.\right)$.Covariance offset hypothesis: the marginal distribution of the two domains is different, that is, $P_{X}^{s}\left(x\right)\neq P_{X}^{t}\left(x\right)$, but the conditional probability distribution of the label is the same, that is, $P_{Y\left|X\right.}^{s}\left(y\left|x\right.\right)=P_{Y\left|X\right.}^{t}\left(y\left|x\right.\right)$ (equivalent to the learning function $g_{s}=g_{t}=g$).

In wireless FL, the typical scenario that requires transfer learning can be summarized into two cases:The channel parameters on one or more links are changed;The channel parameters of a local region are changed.
Whichever case is considered, the above hypotheses are too strong to be satisfied. First, the class imbalance hypothesis requires that the distribution of fingerprints at each location remain the same. Second, the covariance offset hypothesis requires that the position recognition function be the same. Therefore, the transfer learning algorithm based on these two hypotheses is easy to fail in FL.

In addition, the feature-based transfer learning approach assumes the existence of a pair of mapping functions $\left\{\varphi_{s}\left(\cdot \right),\varphi_{t}\left(\cdot \right)\right\}$ that maps features from the source and the target domains to a common latent domain, while the labels remain unchanged. Therefore, the learning function in target domain $g_{t}\left(\cdot \right)$ is approximately replaced by $g_{l}:\varphi_{s}\left(x_{s}\right)\to u_{s}$, the position recognition function trained in the latent domain.

## 4. Transfer Learning for Fingerprinting Localization

### 4.1. Transfer Component Analysis

Transfer Component Analysis (TCA) [27] is one of the most usual feature-based transfer learning methods, whose principle is to adaptive the marginal distribution of the feature. TCA learns the cross-domain transfer components in the reproducing kernel Hilbert space by minimizing the MMD distance between the source domain and the target domain samples after mapping. Let the number of samples in the source domain and target domain be $n_{s}$ and $n_{t}$, respectively, and the MMD distance between them is [32]
(2)$$D\left(X_{s},X_{t}\right)=∥\frac{1}{n_{s}}\sum_{i=1}^{n_{s}}\varphi (x_{i})-\frac{1}{n_{t}}\sum_{j=1}^{n_{t}}\varphi (x_{j})∥,$$
where $X_{s}$ and $X_{t}$ represent the fingerprint sample matrix of source domain and target domain, respectively. However, it is usually highly nonlinear and difficult to directly minimize the MMD distance. The above distance can be converted into kernel function form, so the problem is converted to kernel matrix learning and written as semi-definite program (SDP) in the form of
(3)$$\max_{K≻0}\;tr\left(\mathrm{KL}\right)-\lambda tr\left(K\right),$$
where K is the kernel matrix defined on all data. Let $K_{s,s}$, $K_{t,t}$, and $K_{s,t}$, respectively, represent the Gram matrix defined on the source domain, target domain, and cross-domain data, $K_{ij}=\varphi {\left(x_{i}\right)}^{T}\varphi \left(x_{j}\right)$; then,
(4)$$K=\left[\begin{array}{cc} K_{s,s} & K_{s,t}\\ K_{s,t}^{T} & K_{t,t} \end{array}\right].$$

The elements in matrix L is calculated as
(5)$$L_{ij}=\left\{\begin{array}{cc} \frac{1}{n_{s}^{2}} & x_{i}x_{j}\in X_{s}\\ \frac{1}{n_{t}^{2}} & x_{i}x_{j}\in X_{t}\\ -\frac{1}{n_{s}n_{t}} & otherwise \end{array}\right.$$

The results can be constructed with dimension reduction method, that is, solve the first *m* eigenvalues of ${\left(\mathrm{KLK}+\mu I\right)}^{-1}\mathrm{KHK}$, reducing the computational cost of solving SDP. The TCA method is the base method in feature-based transfer learning, many other methods are extended upon it, such as JDA [28] and BDA [29].

### 4.2. Optimal Transport For Fingerprint Transfer Learning

#### 4.2.1. Basic Method

In this paper, we implemented the optimal transport (OT) method in the transfer learning of FL. Different from the feature-based method (such as TCA), it is assumed that the drift of the domain is caused by an unknown transformation $T:X_{s}\to X_{t}$ from the distribution of source domain to the distribution of target domain. In the FL, the corresponding physical interpretation of the drift may be the change of fingerprint acquisition conditions, changes in environmental parameters, changes in noise conditions, or other unknown processes. Let us say that transformation *T* maintains the condition distribution of the location label in this process, namely,
(6)$$P_{t}\left(u_{t}\left|x_{t}\right.\right)=P_{s}\left(u_{s}\left|T(x_{s})\right.\right).$$

This means that the transformation maintains the information of the position decision function, so the position estimator in the target domain can be approximated by the estimator, $g_{t}\left(T(x_{s})\right)$, trained after the source domain is mapped to the target domain, as shown in the schematic diagram in Figure 1. Then, the knowledge about the location recognition function is transferred from the old radio map to the new one.

From the perspective of probability, *T* transforms the marginal measure $p_{X}^{s}$ of a fingerprint on the source field to the measure of its image, which is represented by $T\#p_{X}^{s}$. It is another measure on $X_{t}$, which satisfies
(7)$$T\#p_{X}^{s}\left(x\right)=p_{X}^{s}\left(T^{-1}\left(x\right)\right).$$

If $T\#p_{X}^{s}=p_{X}^{t}$, T is called a transport map from $p_{X}^{s}$ to $p_{X}^{t}$. Under this assumption, $X_{t}$ comes from the same probability distribution as $T\#p_{X}^{s}$. Therefore, the principle of solving the problem of transfer learning in FL through OT is the same as that in [16]:The probability measures $p_{X}^{s}$ and $p_{X}^{t}$ are estimated using $X_{s}$ and $X_{t}$.Find a transport map T, from $p_{X}^{s}$ to $p_{X}^{t}$.The labeled sample $X_{s}$ is transported with T, and then the target domain estimator $g_{t}\left(\cdot \right)$ is trained with the transformed samples.

The key point is to find the right transport T. OT finds T by minimizing the transport cost C(T):(8)$$C\left(T\right)=\int_{X_{s}}c(x,T\left(x\right))dp_{X}^{s}\left(x\right),$$
where the cost function $c:X_{s}\times X_{t}\to R^{+}$ is a distance function in the feature space X. C(T) can be interpreted as the total energy required to move the fingerprint probabilistic mass $p_{X}^{s}$ to the fingerprint probabilistic mass $T\#p_{X}^{s}$. The solution to the OT problem defined by The Monge problem is
(9)$$\begin{array}{cc} \; & T_{0}=\underset{T}{\arg\min}\;\int_{X_{S}}c(x,T\left(x\right))dp_{X}^{s}\left(x\right)\\ \; & s.t.T\#p_{X}^{s}=p_{X}^{t} \end{array}.$$

The problem in (Equation 9) is combinatorial, and the feasible set of which is nonconvex. Therefore, solving the Monge problem is difficult. Kantorovitch form of OT is a convex relaxed version. Define $\Pi \in P(X_{S}\times X_{T})$ as a set of probabilistic coupling, whose marginal measures are $p_{X}^{s}$ and $p_{X}^{s}$. The Kantorovitch problem requires finding the probabilistic coupling minimizing the following formula,
(10)$$\gamma_{0}=\underset{\gamma \in \Pi }{\arg\min}\;\int_{X_{s}\times X_{t}}c(x_{s},x_{t})d\gamma \left(x_{s},x_{t}\right),$$
where γ can be regarded as the joint probability measure with marginal measure $p_{X}^{s}$ and $p_{X}^{t}$, also known as transport map. The above formula has been proved to be applicable to define the distance between distributions, which is called Wasserstein distance or Earth Mover distance. The Wasserstein distance of order *n* between $p_{X}^{s}$ and $p_{X}^{t}$ is defined as
(11)$$\begin{array}{cc} \; & d_{w,n}\left(p_{X}^{s},p_{X}^{t}\right)\equiv {\left(\underset{\gamma \in \Pi }{\inf}\;\int_{X_{s}\times X_{t}}d(x_{s},x_{t}{)}^{n}d\gamma \left(x_{s},x_{t}\right)\right)}^{\frac{1}{n}}\\ \; & ={\left(\underset{\gamma \in \Pi }{\inf}\;\left(E_{x_{s}∼p_{X}^{s},x_{t}∼p_{X}^{t}}d{\left(x_{s},x_{t}\right)}^{n}\right)\right)}^{\frac{1}{n}} \end{array}$$

Compared with MMD distance defined in formula (Equation 2), Wasserstein distance better describes the contour and detail differences of the two distributions.

As discrete samples are obtained in the actual situation, only the empirical distributions of $p_{X}^{s}$ and $p_{X}^{s}$ can be obtained, denoted as
(12)$$p_{X}^{s}=\sum_{i=1}^{n_{s}}p_{i}^{s}\delta_{x_{i}^{s}},p_{X}^{t}=\sum_{i=1}^{n_{t}}p_{i}^{t}\delta_{x_{i}^{t}},$$
where $\delta_{x_{i}}$ and $p_{i}$ represent the Dirac function and the probabilistic mass at the sample $x_{i}$, respectively. $p_{i}$ belongs to the probabilistic simplex, namely, $\sum_{i}p_{i}=1$. Then, define the probability coupling matrix as follows,
(13)$$B=\left\{\gamma \in {\left(R^{+}\right)}^{n_{s}\times n_{t}}|\gamma 1_{n_{t}}=p_{X}^{s},\gamma^{T}1_{n_{s}}=p_{X}^{t}\right\},$$
where $1_{d}$ is the *d*-dimensional full 1 column vector. The OT of the discrete version of Kantorovitch form can be expressed as
(14)$$\gamma_{0}=\underset{\gamma \in B}{\arg\min}\;{\langle \gamma ,C\rangle }_{F},$$
where ${\langle \cdot ,\cdot \rangle }_{F}$ represents the Frobenius product, C≥0 represents the cost function matrix, and $C\left(i,j\right)=c\left(x_{i}^{s},x_{j}^{t}\right)$ represents the cost required to transport the probability mass from $x_{i}^{s}$ to $x_{j}^{t}$. For simplicity, the Euclid square distance is used as the cost in this paper, that is, $c\left(x_{i}^{s},x_{j}^{t}\right)={∥x_{i}^{s}-x_{j}^{t}∥}_{2}^{2}$.

In general, $\gamma_{0}$ is a sparse matrix containing $n_{s}+n_{t}-1$ nonzero elements at most. (Equation 14) is a linear programming problem, which can be solved by simplex method [33]; however, the complexity is high. The OT regularized with entropy adds an entropy regularized term $\Omega s\left(\gamma \right)$, namely,
(15)$$<\gamma_{0}{>}_{\mathrm{Sin}k.}=\underset{\gamma \in B}{\arg\min}\;{\langle \gamma ,C\rangle }_{F}+\lambda_{e}\Omega_{e}\left(\gamma \right),$$
where $\Omega_{e}\left(\gamma \right)=\sum_{i,j}\gamma (i,j)\log\gamma (i,j)$ is the negative entropy of γ, and $\lambda_{e}$ represents the corresponding regularization coefficient. By adding the entropy of γ, a smoother transport diagram can be obtained, and a more efficient algorithm has been derived, called Sinkhorn-knopp [34].

After $\gamma_{0}$ is solved, barycentric matching [35] can obtain the mapping of all source domain samples in the target domain:(16)$${\hat{X}}_{s}=n_{s}\gamma_{0}X_{t}.$$

The purpose of the OT transfer learning is to correctly recover the transport graph from the data distribution in the source domain to the data distribution in the target domain, and what kind of transformation it can recover has not been proved theoretically. However, it has been proved that the affine transformation of discrete distributions can be recovered lossless [30].

#### 4.2.2. Laplacian Regularization

In FL, fingerprints that are intuitively “close” in the source domain should also be “close” when transported to the target domain, and vice versa. Let $\;{\hat{x}}_{i}^{s}$ represent the value after the source domain sample $x_{i}^{s}$ is mapped to the target domain, and the Laplacian regularization term is introduced:(17)$$\Omega_{l}\left(\gamma \right)=\alpha \cdot \frac{1}{n_{s}^{2}}\sum_{i,j}S_{s}\left(i,j\right)∥{\hat{x}}_{i}^{s}-{\hat{x}}_{j}^{s}{∥}_{2}^{2}+\left(1-\alpha \right)\cdot \frac{1}{n_{t}^{2}}\sum_{i,j}S_{t}\left(i,j\right)∥{\hat{x}}_{i}^{t}-\;{\hat{x}}_{j}^{t}{∥}_{2}^{2},$$
where $S_{s}\left(i,j\right)\geq 0$ is the element of sample similarity matrix $S_{s}$ in the source domain, and $S_{t}$ is similar. α is a super parameter, which represents the importance factor of Laplacian regularization in the source domain. When the marginal distribution is uniform, the above formula can be further simplified by formula (Equation 16):(18)$$\Omega_{l}\left(\gamma \right)=\alpha Tr\left(X_{t}^{T}\gamma^{T}L_{s}\gamma X_{t}\right)+\left(1-\alpha \right)Tr\left(X_{s}^{T}\gamma L_{t}\gamma^{T}X_{s}\right),$$
where $L_{s}=diag\left(S_{s}1\right)-S_{s}$ is the Laplacian matrix associated with the graph $S_{s}$, similarly, $L_{t}=diag\left(S_{t}1\right)-S_{t}$. When using Laplacian regularization, we solve the following problem,
(19)$$<\gamma_{0}{>}_{\mathrm{EMDL}}=\underset{\gamma \in B}{\arg\min}\;{\langle \gamma ,C\rangle }_{F}+\lambda_{l}\Omega_{l}\left(\gamma \right),$$
where $\lambda_{l}$ represents the coefficient of the Laplacian regularization. Then, the subsequent matching process goes the same as the Sinkhorn algorithm.

#### 4.2.3. Joint Estimation of Transport Map and Transformation Function

The transport map is responsible for transporting the empirical probabilistic mass from the source domain to the target domain, or vice versa. The algorithm gets the transport map of probability density, not a transformation function. Jointly learning the transport map and transformation function makes the learner better extend to unknown samples, which is known as out-of-sample case [36]. The cost function of joint transport and transformation estimation is
(20)$$<\gamma_{0},T_{0}{>}_{M.\;\mathrm{OT}}=\underset{\gamma \in B,T\in H}{\arg\;\min}\;\frac{1}{n_{s}m}{∥T\left(X_{s}\right)-n_{s}\gamma X_{t}∥}_{F}^{2}+\frac{\lambda_{\gamma }}{\max(C)}{\langle \gamma ,C\rangle }_{F}+\frac{\lambda_{T}}{m^{2}}R\left(T\right),$$
where $R\left(\cdot \right)$ is the regularization term related to the transformation *T*; $\lambda_{\gamma }$ and $\lambda_{T}$ are regularization coefficients. The hypothesis space of the transformation *T*, H, can be either a linear or nonlinear function space, and we adopted a linear function in this paper.

#### 4.2.4. Data Preprocessing and Optimization Algorithm

In our experiments, the source domain and target domain data are both normalized by subtracting the mean value and dividing by the variance of the data. Data preprocessing is rarely mentioned in former transfer learning studies, but it significantly impacts performance. This is because data preprocessing can reduce the difference of mean value and variance between two domains to a certain extent, which is similar to the effect of ”transfer”.

For the optimization of the Laplacian regularization version in Formula (Equation 19), we adopt the Generalized Conditional Gradient (GCG) algorithm [37] to solve the optimization of the OT problem in this paper. The regularization in (Equation 19) is strictly convex, so the objective function can reach the minimum on B. Specifically, using f(γ) represents the objective function in formula (Equation 19), the method iterates the steps below until convergence:(21)$$\begin{array}{cc} \; & {\tilde{\gamma }}^{l+1}=\underset{\gamma \in B}{\arg\;\min}\langle \gamma ,\nabla f(\gamma^{l})\rangle ,\\ \; & \gamma^{l+1}=\gamma^{l}+\tau^{l}\left({\tilde{\gamma }}^{l+1}-\gamma^{l}\right), \end{array}$$
where $\tau^{l}$ is obtained by linear search. For the optimization of the joint matching algorithm version in Formula (Equation 20), the block-coordinate descent (BCD) [38] method is used. The idea is to alternatively optimize λ and *T*.

These optimization algorithms are available on a public website [39] that readers can refer to.

## 5. Wireless Fingerprint Channel Model

In this paper, RSS fingerprint characteristics are taken as an example, assuming that there are *m* APs in the interesting region and the region is a 2-dimensional Euclidean space. The coordinate of the *i*-th AP is $a_{i}$, i=1,…,m. The APs transmit wireless signals at a certain power. The fingerprint at the coordinate u in the source domain is represented by $x_{s}\left(u\right)$, which is composed of the received power from all APs, namely, $x_{s}\left(u\right)=\left[x_{1}^{s}\left(u\right),x_{2}^{s}\left(u\right),\ldots ,x_{m}^{s}\left(u\right)\right]$. In the target domain, the channel link parameters change, and the fingerprint at coordinate u is represented by $x_{t}\left(u\right)$. The commonly used model of RSS fingerprints are free-space model, multi-wall model, and ray tracing [40]. In this paper, the free space model and multi-wall model are used to simulate the distribution changes in fingerprint transfer.

### 5.1. Free Space Loss Model

In the free space loss model, it is assumed that the received power is calculated over a long period of time without considering the small-scale fading of the channel. Assuming that the receiving power (unit mW) follows a lognormal distribution, the fingerprint component of the mobile device located at u from the *i*-th AP is [41]
(22)$$x_{i}\left(u\right)=P_{i}-10\cdot \alpha_{i}\log_{10}\left(∥a_{i}-u∥/d\;\right)+N_{i},$$
where $P_{i}$ represents the received power of the *i*-th AP at the reference distance *d* (usually 1 m), with unit dBm; $\alpha_{i}$ is the path loss exponent of the *i*-th AP; and $N_{i}$ is the link noise of the *i*-th AP, which obeys a Gaussian white noise with variance $\sigma_{i}^{2}$. Assuming that AP is independent of each other, the conditional probability density function of the location fingerprint is
(23)$$P\left(x\left|u\right.\right)=\prod_{i=1}^{n}\frac{1}{\sqrt{2\pi \sigma_{i}^{2}}}\exp\left[-\frac{{\left(x_{i}-\left(P_{i}-10\cdot \alpha_{i}\log_{10}\left(∥a_{i}-u∥/d\;\right)\right)\right)}^{2}}{2\sigma_{i}^{2}}\right].$$

The free space loss model is a signal propagation model in an ideal environment. In this paper, the free space model is used to model the fingerprint changes when the link parameters of the channel change. Suppose the link noise of all AP is equal, that is, $N_{1}=N_{2}=\cdots =N_{m}=N$, and remains unchanged. When the link parameters change from $\theta_{s}={\left[P_{1}^{s},P_{2}^{s},\ldots ,P_{m}^{s},\alpha_{1}^{s},\alpha_{2}^{s},\ldots ,\alpha_{m}^{s}\right]}^{T}$ to $\theta_{t}={\left[P_{1}^{t},P_{2}^{t},\ldots ,P_{m}^{t},\alpha_{1}^{t},\alpha_{2}^{t},\ldots ,\alpha_{m}^{t}\right]}^{T}$, the conditional probability of location fingerprint will change from $P_{s}\left(x\left|u;\theta_{s}\right.\right)$ to $P_{t}\left(x\left|u;\theta_{t}\right.\right)$.

### 5.2. Multi-Wall Model

The multi-wall model is an extension of the single-slope loss model, including an additional attenuation term, which is caused by the loss of direct path between transmitter and receiver encounters the wall and door [42]. In the multi-wall model, Formula (Equation 21) is rewritten as
(24)$$x_{i}\left(u\right)=P_{i}-10\cdot \alpha_{i}\log_{10}\left(∥a_{i}-u∥/d\;\right)-M_{w}+N_{i},$$
where the additional attenuation term can be expressed as
(25)$$M_{w}=l_{c}+\sum_{i=1}^{I}k_{wi}l_{i}+\sum_{n=1}^{N_{d}}\chi_{n}l_{d}+\sum_{n=1}^{N_{fd}}\lambda_{n}l_{fd},$$
where $k_{wi}$ represents the number of penetrable wall of type *i*, $l_{i}$ is the corresponding loss of signals passing through it; $N_{d}$ and $N_{fd}$ are the number of ordinary doors and fire doors passing through the direct path, respectively; $l_{d}$ and $l_{fd}$ are losses corresponding to signals passing through ordinary doors and fire doors, respectively; and $\chi_{n}$/$\lambda_{n}$ is a binary variable, indicating the state of the *n*-th ordinary door/fire door.

The multi-wall model considers the attenuation of the signal after passing through the wall, which can be used to simulate the RSS fingerprint under different indoor structures. In the experiment part, the multi-wall model is used to model the transfer learning when the local area’s channel parameters change. Compared with the source domain, some wall structures of the target domain are changed.

## 6. Experiments

To verify the performance of OT-based transfer learning in FL, two models described in Section 5 are used for numerical simulation, and the performance of the algorithm is also verified with the public data set. First, we use the free space channel model to simulate the transfer scenario of RSS fingerprint when the radio link parameters change. Second, the indoor multi-wall model was used to simulate RSS fingerprint transfer’s learning scene when the environmental parameters of a local area changed. Finally, the performance of the algorithm is further verified by using the publicly measured data set. One of the performance evaluation indicators used in this paper is the average positioning error (AE), and its calculation formula is
(26)$$AverageError=\frac{1}{n_{t}}\sum_{j=1}^{n_{t}}\sqrt{{∥{\hat{u}}_{j}-u_{j}∥}_{2}},$$
where ${\hat{u}}_{j}$ and $u_{j}$, respectively, represent the estimated and real coordinates of the *j*-th sample in the target domain test set, and $n_{t}$ represents the total number of the test samples. The cumulative error value of 50% and 80% was used as additional evaluation indicators.

### 6.1. Free Space Channel Model RSS FL Transfer Learning Simulation

The simulation was set as a 1-d positioning scene, with range [0,10] and 2 APs located at −1 and 11, respectively. Using the model described in Section 5.1, the channel parameters, $P_{i}$ and $\alpha_{i}$, are changed to simulating the change in the radio link parameters. The parameters of the source domain and target domain are shown in Table 1. In the source domain, the positioning area is divided into 10 grid points, and the center of each grid serves as the real label of the position. Ten samples are randomly generated in each grid as the training set. In the target domain, 1000 samples were randomly collected as test sets. The standard deviation of the noise was set at 2 dBm.

The TCA method and OT linear map [43] were used to transfer the samples of the above simulation scenarios. The left side of Figure 2 shows the normalized fingerprint samples in the source domain and the target domain. The two axes, respectively, represent the two features, and the size of the sample point represents the relative value of the real coordinate values. Accordingly, the right side of Figure 2 shows the results after the source domain and the target domain are mapped to the latent domain through TCA transformation. Figure 3 shows the fitted distributions of the two features on the left and the fitted distributions after mapping to the latent domain through TCA transformation on the right.

It can be observed from Figure 2 and Figure 3 that TCA transformation fails to match the distributions of the two domains. The samples’ variance in the source domain is large, while it is small in the target domain. This is still the case after mapping to the latent domain. We know that the objective of TCA is to minimize the mean difference of the mapped samples. As the samples have been normalized, the mean difference between the two domains is relatively small. Therefore, TCA does not significantly improve it. As the cost function does not constrain the variance, the variance between the two distributions is still massive after TCA transfer.

Figure 4 shows the changes in the fingerprint samples in the data domain before and after the OT-based transfer learning. The left part of Figure 4 shows the samples in the fingerprint space from the source domain, the target domain, and the target domain mapped from the source domain through OT. The middle and the right part are the fitted distributions of the two features in different domains.

It can be observed from Figure 4 that the samples mapping from the source domain to the target domain with OT have a higher matching degree with the sample distribution in the target domain, whatever the mean, the variance, or the contour. We know that OT is to transport the source domain’s distribution to the distribution of the target domain under the principle of minimum cost, so the two distributions are matched better.

According to the Transfer Learning theory, the generalization error bound of transfer learning is related to the distance between the distributions of the two domains, and the smaller the distance is, the lower bound will be reached [44]. It is observed from the above simulation that the distributions obtained by OT match better than what by TCA. Moreover, Figure 5 shows the relationship between the average test error with changing the noise level without transfer learning, TCA learning, and OT learning. It can be observed from Figure 5 that the transfer performance using OT is the best, and the advantage gradually decreases with the increase of noise level. Besides, TCA does not improve the target domain location performance but rather weakens. From the fitted distribution of Figure 3 and Figure 4, it is obvious that the positioning performance has a great correlation with the degree of distribution coincidence.

### 6.2. Multi-Wall Model of RSS FL Transfer Learning Simulation

We use the indoor multi-wall model to simulate the transfer learning algorithm’s performance when the local propagation environment changes. Figure 6 shows the heat maps of the power obtained by simulating two environments with 6 APs placed at the same relative locations. Roughly, the two environments have the same layout, and the area is about 50∗20 m, but there are fewer walls in Environment 1. The simulation data of Environment 1 are taken as samples in the source domain. Some walls are added in Environment 2 to simulate the local changes compared with Environment 1. The simulation data of Environment 2 is taken as samples in the target domain. In both environments, 7200 fingerprint samples were collected at uniform and the same locations, without noise. It can be observed from Figure 6 that there are some local differences in the power energy due to local layout changes.

Figure 7 shows a heat map of the power mapped from source to latent domain by TCA transformation on the left and from target to latent domain on the right. The data are preprocessed by normalization before the transformation. It can be observed that the heat map on the right is still locally different from that on the left.

Figure 8 shows the power heat map of the source domain on the left, the target domain in the middle, and the source domain after mapping to the target domain using OT on the right. It can be seen that OT makes the heat map mapped from the source domain more similar to that in the target domain. However, note that the heat map is the superposition of all APs, so it is not possible to ultimately determine whether the two are similar from this figure alone.

Finally, the relationship between the algorithms’ average test error and the noise-level is shown in Figure 9. The curve in Figure 9 has a similar trend with that in Figure 5. This indicates that OT plays a positive transfer role in both models, while the TCA method appears to negative transfer [40] in both cases.

### 6.3. Measured Data Experiment

In this section, we use the recent publicly available measured data set [45] to test the performance of transfer learning based on OT and typical feature-based transfer learning algorithms in FL. The data set was collected at the same locations in a library over 15 natural months, using the same device each month. A total of 448 APs were detected in the whole data set. However, due to the long period of sample collection, the number of APs detected every month was different, especially since the 12th month, which had a significant change. In this experiment, the first month’s data were taken as the sample set of the source domain, with a total of 8640 samples. The mean value of 6 samples in a continuous period and at the same location was taken, and 1440 training samples were finally obtained. The remaining data of each month is taken as the target domain sample set, which contains 6 data sets each month: one of them is taken as the validation set, containing 576 samples; 3 of them are taken as the test set 1, containing 1728 samples; the remaining two are taken as the test set 2, containing 1392 samples. This experiment focuses on the time transfer of location fingerprints, namely, training the model with the first month’s data and then using the transfer learning algorithm to transfer the model to the unlabeled samples of the remaining months.

The validation set is used for choosing the super parameters. For each transfer learning algorithm, the grid searching method is used to select the super parameters that make the validation set perform best. In the transductive setting, the test set 1 is used as the target domain samples and test samples simultaneously. In the out-of-sample setting, the transfer learning algorithm uses the test set 1 as the target domain samples, but the test set 2 is used for testing. In this paper, the classical TCA transfer learning algorithm and its extension method BDA are selected as the comparison algorithms. In addition, the performance without transfer learning was also included in the comparison (represented by Without). The other three methods are based on the OT method: EMDL is the OT method with Laplacian regularization (described in Section 4.2.2), Sinkhorn is the OT method with entropy regularization (described in Section 4.2.1), and Map. OT is the joint estimation method (described in Section 4.2.3). The average positioning error of month 2 to 15 with different algorithms is shown in Figure 10, under the transductive setting on the left and out-of-sample setting on the right. It can be observed from the figure that traditional transfer learning methods (TCA and BDA) showed no significant performance improvement during months 2 to 10 under both settings. While under the transductive setting the average error of transfer learning based on OT was reduced by about 10% compared with not using any transfer algorithms. All transfer learning methods improved in accuracy after month 10, and the OT-based method performs even better under both settings. When the cumulative error distribution function values are at 50% and 80% for each algorithm, the corresponding error values (represented by C.5 and C.8, respectively) are shown in Table 2 and Table 3. We can draw similar conclusions from these results.

The results from the real data set have a slight difference from the simulated data. Nevertheless, they all show the superior performance of the OT-based methods.

### 6.4. Super Parameters

In this section, we explain how the super parameters in the algorithm are selected and how they affect the average positioning error. In the Sinkhorn algorithm, there is a super parameter $\lambda_{e}$, namely, the entropy regularization coefficient. We empirically select $\lambda_{e}$ that minimizes the validation error, as shown in Figure 11. For space reasons, we only show the result in the 12th month, still using the 1st month data as the source domain data. It can be seen from the figure that when $\lambda_{e}$ is set at 0, the error is large, and when it is set at 20, the error reaches the minimum value, indicating that moderate entropy regularization plays a role in improving the performance.

There are two super parameters in the EMDL algorithm, $\lambda_{l}$ and α, which represent the regular coefficient of Laplacian and the proportion of the importance of the source domain data. The M. OT algorithm also has two super parameters, namely, $\lambda_{\gamma }$ and $\lambda_{T}$, which represent the regularization coefficient of transport cost and the regularization coefficient of the matching function. Similarly, in Figure 12 and Figure 13, we, respectively, show the variation of the validation error when taking different super parameters in the 12th month.

As you can see from Figure 12, in the EMDL algorithm, $\lambda_{l}$ and α both need to be set to a larger value to achieve better performance. This indicates that Laplacian regularization has improved performance and that the source domain data is of greater importance. As can be seen from Figure 13, the selection of $\lambda_{T}$ in the M. OT algorithm is more important than $\lambda_{\gamma }$ to the performance, but, in general, they have little impact on the positioning mean error.

For these three algorithms, we observe that the optimal super parameter value of each month is almost unchanged, which is not shown in the paper due to the limited space. This shows that our algorithm has good robustness in super parameter selection.

## 7. Discussion

The problem of transfer learning is closely related to the distance metric of data distribution, and OT is an important tool to study the distance of data distribution. The distance derived from OT has many good properties. In this paper, the OT method is applied to the transfer learning of FL. The simulation and measured fingerprint data of different fingerprint models show that the transfer learning method based on the OT method has better transfer performance than the traditional TCA method. We find that this is related to the way they define the distance measure of the empirical distribution. The latter’s MMD distance is the most commonly used distance metric in transfer learning, but it only describes the mean value of the distribution, while the Wasserstein distance used by the former gives a good description of the details of the distribution. The latest transfer learning review article does not cover the methods based on OT, which we believe should receive wider attention. This paper analyzes and conducts experiments under two different channel models for the transfer learning problem in FL. We find that the traditional method has a negative transfer effect in the experiment, while the OT method can achieve positive transfer, indicating that the occurrence of negative transfer is related to the algorithm. The experimental study in this paper causes us to think about the following theoretical questions.
What conditions the location fingerprints can be positively transferred under?How good the generalization bound can be reached in the transfer learning of FL?What causes the difference between the simulation model and the real data?
These questions will be further studied in our following work.

## Figures and Tables

**Figure 1 sensors-20-06994-f001:**
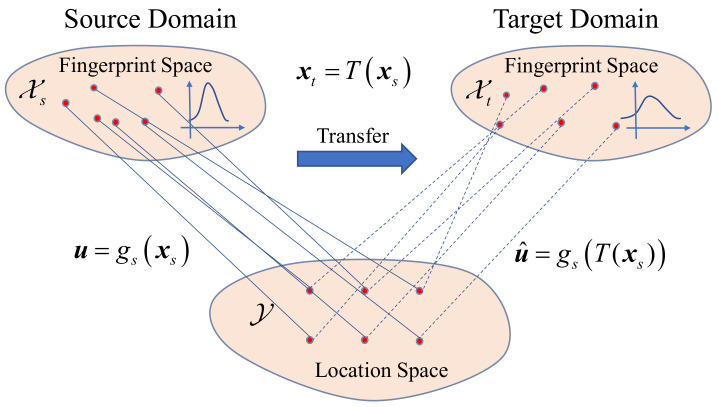
Schematic diagram of the transfer learning in FL based on OT.

**Figure 2 sensors-20-06994-f002:**
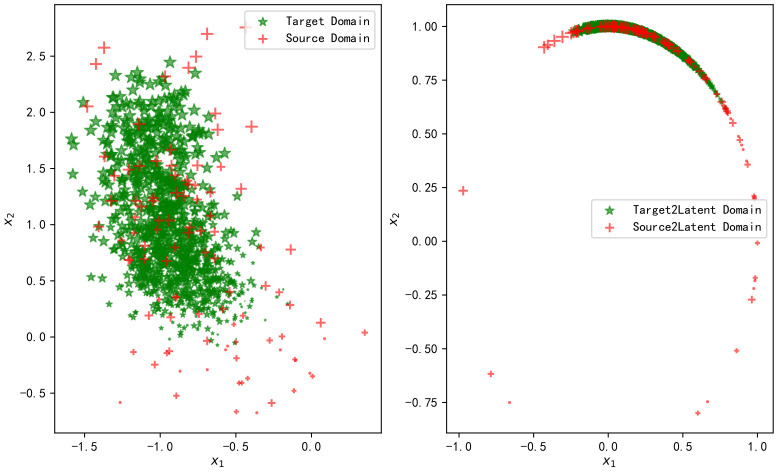
Fingerprint samples in source and target domains (**left**) after TCA transformation to the latent space (**right**).

**Figure 3 sensors-20-06994-f003:**
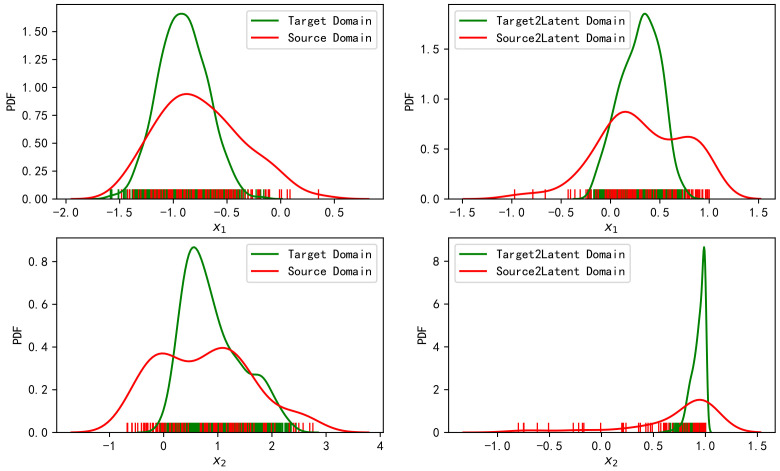
Fitted distributions of features in the source domain and target domain (**left**), and the fitted distributions of features after TCA transformation to the latent space (**right**).

**Figure 4 sensors-20-06994-f004:**
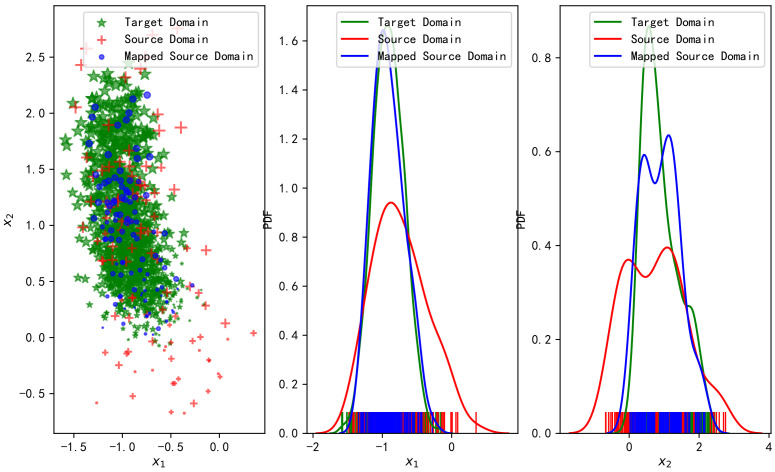
Fingerprint samples in the source domain, in the target domain, and after mapping from the source domain to the target domain with OT (**left**), the corresponding fitted distributions of features (**center** and **right**).

**Figure 5 sensors-20-06994-f005:**
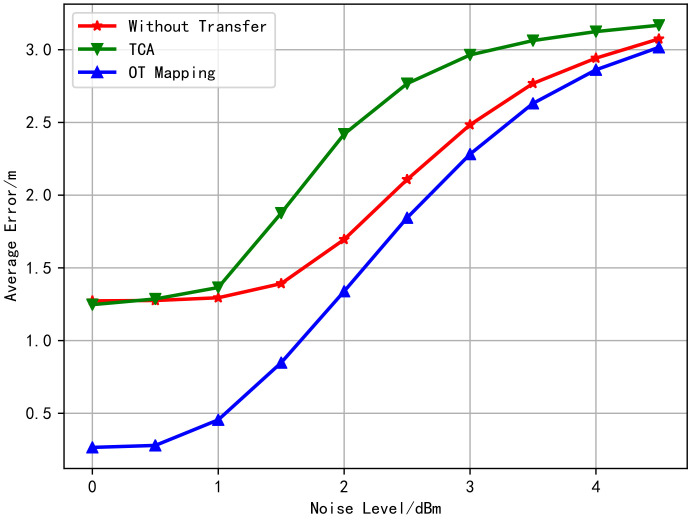
Comparison of the average positioning error of the transfer learning algorithms (using free space loss model).

**Figure 6 sensors-20-06994-f006:**
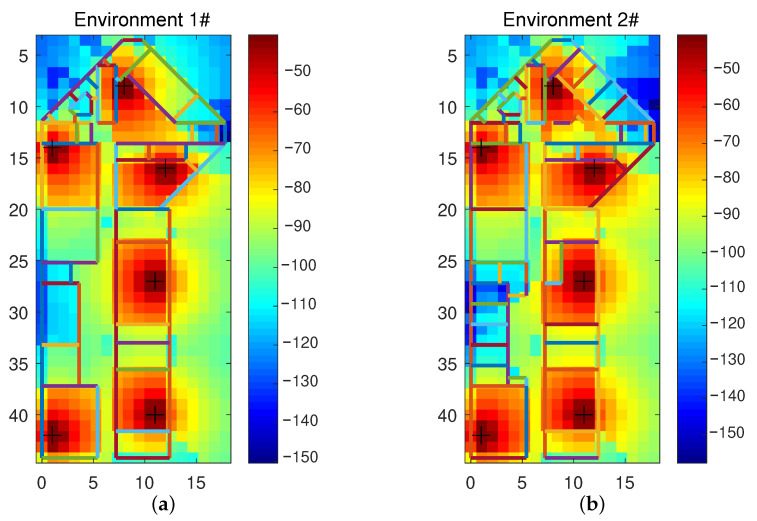
The layout (unit: meter) and the power heat maps (unit: dBm) of simulated indoor multi-wall model: (**a**) Environment 1/the source domain; (**b**) Environment 2/the target domain.

**Figure 7 sensors-20-06994-f007:**
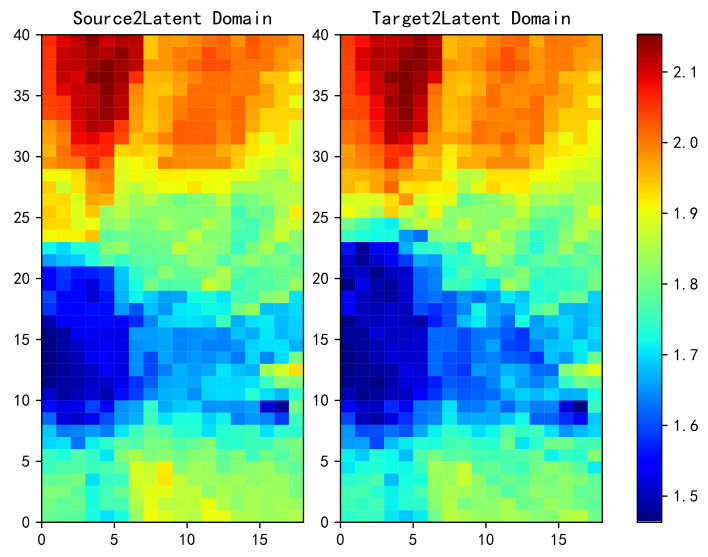
Power and heat graphs from the source domain (**left**) and the target domain (**right**) to latent domain obtained by TCA transformation.

**Figure 8 sensors-20-06994-f008:**
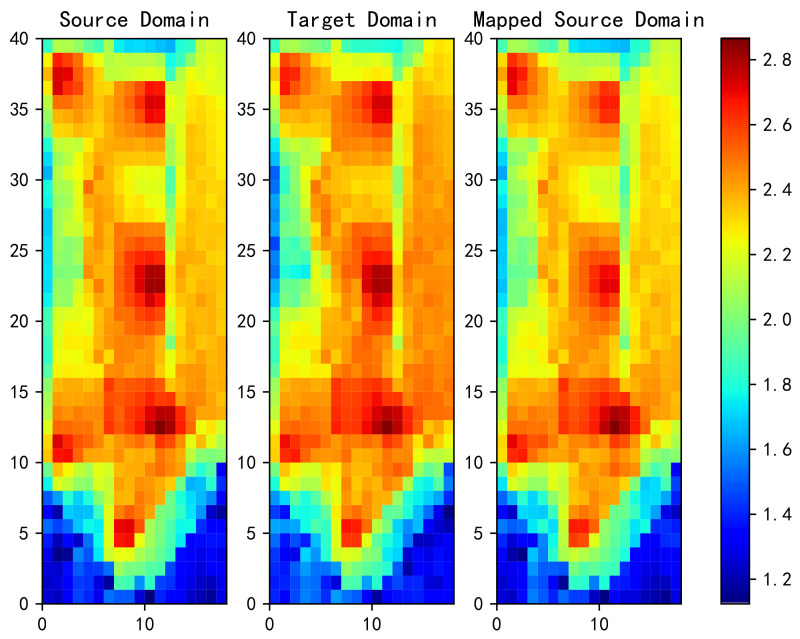
Power heat map in the source domain (**left**), power heat map in the target domain (**middle**), and power heat map after the source domain is mapped to the target domain by OT method.

**Figure 9 sensors-20-06994-f009:**
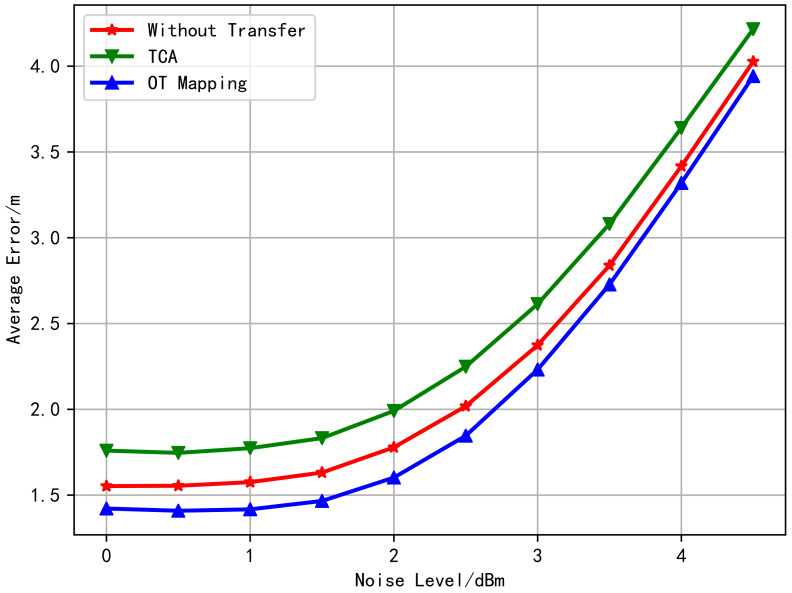
Comparison of the average positioning error of the transfer learning algorithms (using multi-wall model).

**Figure 10 sensors-20-06994-f010:**
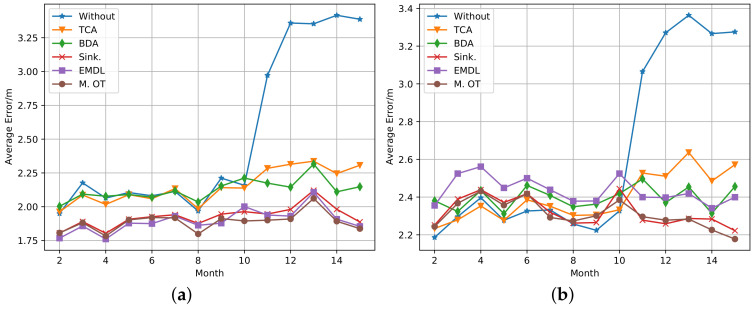
The performance of the transfer learning algorithms on the measured FL data set: (**a**) transductive setting and (**b**) out-of-sample setting.

**Figure 11 sensors-20-06994-f011:**
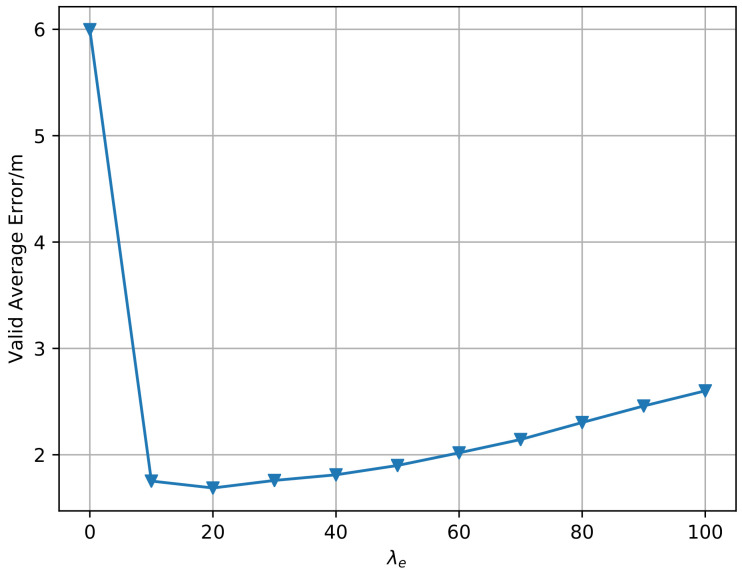
Validation error when taking different super parameters in the 12th month (Sinkhorn).

**Figure 12 sensors-20-06994-f012:**
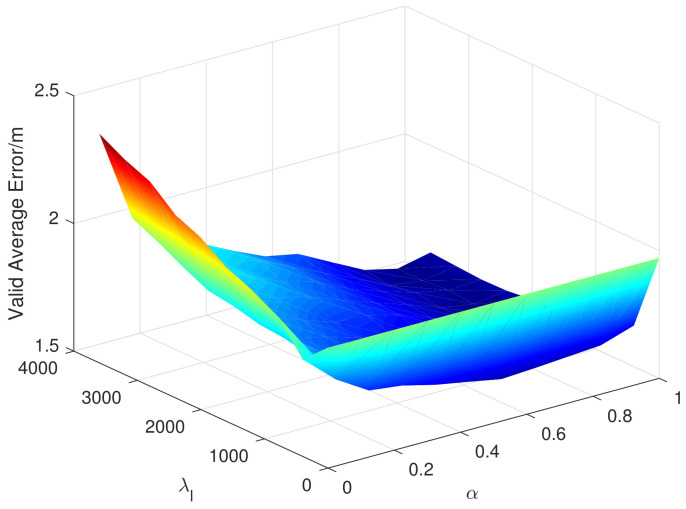
Validation error when taking different super-parameters in the 12th month (EMDL).

**Figure 13 sensors-20-06994-f013:**
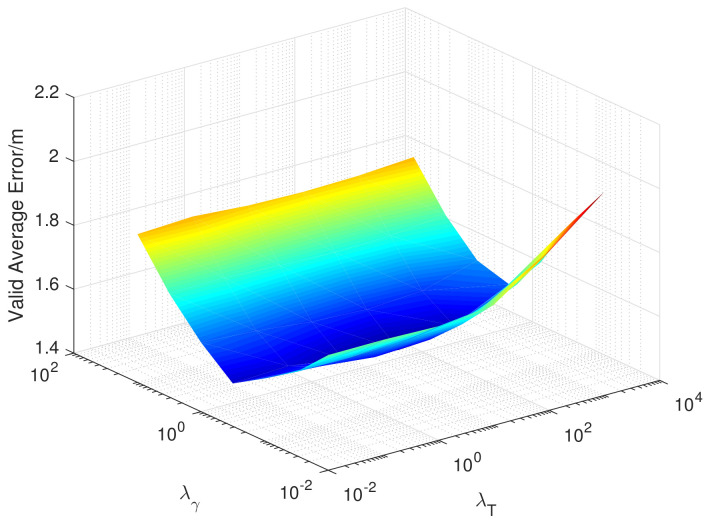
Validation error when taking different super-parameters in the 12th month (M. OT).

**Table 1 sensors-20-06994-t001:** Channel model parameters change in the source domain and the target domain.

AP	AP1		AP2	
	$P_{1}$/dBm	$\alpha_{1}$	$P_{2}$/dBm	$\alpha_{2}$
source domain	50	1	80	3
target domain	40	1	100	4

**Table 2 sensors-20-06994-t002:** The performance of the transfer learning algorithm on the measured FL data set (transductive setting).

Month		02	03	04	05	06	07	08	09	10	11	12	13	14	15
C.5	Without	1.59	1.67	1.72	1.75	1.76	1.76	1.61	1.79	1.76	2.76	3.29	3.23	3.36	3.32
	TCA	1.63	1.63	1.63	1.75	1.75	1.76	1.61	1.79	1.76	2.01	1.99	1.97	1.91	1.99
	BDA	1.69	1.68	1.76	1.76	1.73	1.78	1.67	1.79	1.88	1.89	1.76	1.95	1.76	1.79
	EMDL	**1.42**	**1.45**	**1.42**	**1.45**	**1.47**	**1.48**	**1.45**	**1.54**	**1.48**	**1.6**	**1.49**	**1.75**	**1.47**	**1.48**
	Sink.	**1.34**	**1.34**	**1.34**	**1.43**	**1.44**	**1.46**	**1.36**	**1.44**	**1.47**	**1.49**	**1.45**	**1.54**	**1.34**	**1.43**
	M. OT	**1.44**	**1.44**	**1.44**	**1.45**	**1.5**	**1.47**	**1.42**	**1.47**	**1.45**	**1.5**	**1.45**	**1.61**	**1.42**	**1.44**
C.8	Without	2.93	3.24	3.08	3.2	3.07	3.2	**2.78**	3.2	3.19	4.3	4.6	4.67	4.64	4.57
	TCA	2.98	3.18	3.04	3.19	3.1	3.21	2.96	3.19	3.15	3.39	3.39	3.51	3.24	3.39
	BDA	3.08	3.2	3.08	3.19	3.09	3.12	2.95	3.19	3.2	3.19	3.23	3.51	3.15	3.16
	EMDL	**2.76**	**2.82**	**2.72**	**2.9**	**2.89**	**2.85**	**2.78**	**2.86**	**2.96**	**2.9**	**3.08**	**3.39**	**3.04**	**2.78**
	Sink.	**2.7**	**2.81**	**2.72**	**2.78**	**2.76**	**2.95**	**2.78**	**2.78**	**2.98**	**3**	**2.98**	**3.46**	**2.86**	**2.76**
	M. OT	**2.73**	**2.85**	**2.64**	**2.81**	**2.96**	**2.81**	**2.64**	**2.78**	**2.89**	**2.81**	**2.92**	**3.24**	**2.82**	**2.73**

**Table 3 sensors-20-06994-t003:** The performance of the transfer learning algorithm on the measured FL data set (out of sample setting).

Month		02	03	04	05	06	07	08	09	10	11	12	13	14	15
C.5	Without	**1.99**	**2.06**	2.2	**2.06**	**2.19**	**2.19**	**1.99**	**1.97**	**2.15**	2.83	3.29	3.29	3.25	3.25
	TCA	**2.02**	**1.97**	2.19	**2.08**	**2.22**	**2.19**	2.06	2.16	**2.15**	2.31	2.22	2.37	2.22	2.26
	BDA	2.06	**2.06**	**2.15**	**2.08**	**2.22**	2.2	2.16	2.19	**2.16**	2.31	**2.19**	**2.16**	**2.15**	**2.19**
	EMDL	**2.02**	2.19	**2.16**	2.19	**2.2**	**2.15**	**2.02**	**2.02**	2.2	**2.08**	**1.9**	**1.99**	**2.02**	**1.97**
	Sink.	2.2	2.26	2.26	2.22	2.26	2.22	2.19	2.19	2.22	**2.2**	**2.19**	2.2	2.19	**2.19**
	M. OT	2.08	2.19	**2.16**	2.19	**2.22**	**2.06**	**1.97**	**2.15**	2.2	**2.15**	**1.97**	**2.06**	**2.02**	**1.9**
C.8	Without	**3.12**	**3.33**	**3.47**	**3.31**	**3.21**	**3.36**	**3.22**	**3.2**	**3.25**	4.35	4.49	4.73	4.66	4.64
	TCA	**3.21**	**3.32**	**3.51**	**3.32**	**3.4**	3.51	3.4	**3.36**	**3.4**	3.62	3.65	3.95	3.61	3.78
	BDA	3.58	**3.48**	3.67	3.43	3.55	3.52	3.48	3.52	3.56	3.62	**3.51**	3.67	**3.36**	3.65
	EMDL	3.25	3.51	**3.6**	**3.34**	3.48	**3.47**	**3.21**	**3.25**	3.6	**3.22**	**3.39**	**3.39**	3.48	**3.2**
	Sink.	3.43	3.67	3.82	3.51	3.55	3.52	3.47	3.51	3.75	**3.47**	3.55	**3.58**	**3.47**	**3.47**
	M. OT	**3.2**	3.51	**3.6**	3.4	**3.47**	**3.25**	**3.32**	**3.36**	**3.55**	**3.25**	**3.51**	**3.4**	**3.22**	**3.09**

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
