# Peer review of "Transfer Learning for Wireless Fingerprinting Localization Based on Optimal Transport"

_sensors, 2020, doi:10.3390/s20236994_

Round 1
Reviewer 1 Report
The paper presents an interesting novel method for localization fingerprinting based on machine learning. The paper can be accepted after consideration of the following minor commenst:
The introduction just list technologies for indoor positioning. The authors should discuss the relevant methods in respective to the paper.
What are super parameters? Please clarify and discuss.
Line 139: reference is missing [?]
Title 5.2 should be Multi-wall Model
Line 343: OTation method?
References are partly incomplete, see e.g. 14 and 17
Reference 32: authors missing
Reviewer 2 Report
Authors present an improvement to the wireless fingerprinting localization by using transfer learning. From an indoor positioning, which is a very current topic, point of view, transfer learning could solve some of the inherent problems of fingerprinting-based localization such as the change of wireless stations parameters. The idea is not new, as shown in the relevant and current bibliography in the literature review part of the article, however authors propose to use optimal transport for transfer learning, which is innovative and compares favorably to existing methods.
The article is of good quality, methods are presented in great detail and the experimentation is relevant and supports the hypothesis. Can be published in this form. The only comment is that section 4.2.4 could be expanded and that the discussion could at least hypothesize on the observed differences between measurements based on real and simulated data.
Reviewer 3 Report
The authors present an interesting paper about wireless fingerprinting localization based on optimal transport by tranfer learning from old scenary to new ones. However, it is perhaps, recomended that the authors make a few improvements, namely:
1.- Although the text is pleasant to read, a review of the English writing is recommended. Also, authors should pay attention to some typos, such as: page 4/19 "be succeed [?]". It is a typo, or a reference is missing?
2.- The presentation, motivation and justification of the problem, in general, is well characterized and deescribed. However, with respect to the test/simulation part some questions remain, such as: how were the several data sets generated? Authors use as test condictions two methods, free space model and the multi-wall model, does the generated data take into account the differences between the models? Authors also take into account that, typically, the RSS values of the APs are not stable either spatially or temporally? This is not clear from reading the paper.
Finally, congratulations on your work
